# SELF-CONSISTENT LEARNING: COOPERATION BETWEEN GENERATORS AND DISCRIMINATORS

## ABSTRACT

Using generated data to improve the performance of downstream discriminative models has recently gained popularity due to the great development of pre-trained language models. In most previous studies, generative models and discriminative models are trained separately and thus could not adapt to any changes in each other. As a result, the generated samples can easily deviate from the real data distribution, while the improvement of the discriminative model quickly reaches saturation. Generative adversarial networks (GANs) train generative models via an adversarial process with discriminative models to achieve joint training. However, the training of standard GANs is notoriously unstable and often falls short of convergence. In this paper, to address these issues, we propose a *self-consistent learning* framework, in which a discriminator and a generator are cooperatively trained in a closed-loop form. The discriminator and the generator enhance each other during multiple rounds of alternating training until a scoring consensus is reached. This framework proves to be easy to train and free from instabilities such as mode collapse and non-convergence. Extensive experiments on sentence semantic matching demonstrate the effectiveness of the proposed framework: the discriminator achieves 10+ AP of improvement on the zero-shot setting and new state-of-the-art performance on the full-data setting.

## 1 INTRODUCTION

The advance of Pre-trained Language Models (PLMs) (Brown et al., 2020; Chowdhery et al., 2022) has substantially improved the performance of deep neural networks across a variety of Natural Language Processing (NLP) tasks. Various language models, based on the Transformer (Vaswani et al., 2017) architecture, have been proposed, leading to state-of-the-art (SOTA) performance on the fundamental discrimination tasks. These models are first trained with self-supervised training objectives (e.g., predicting masked tokens according to surrounding tokens) on massive unlabeled text data, then fine-tuned on annotated data to adapt to downstream tasks of interest. However, annotated data is usually limited to a wide range of downstream tasks, which results in overfitting and a lack of generalization to unseen data.

One straightforward way to deal with this data scarcity problem is data augmentation (Xie et al., 2020), and incorporating generative models to perform data augmentation has been widely adopted recently (Carlini et al., 2021; Gangal et al., 2022). Despite its popularity, the generated text can easily deviate from the real data distribution without exploiting any of the signals passed back from the discrimination task. In previous studies, generative data augmentation and discrimination have been well studied as separate problems, but it is less clear how these two can be leveraged in one framework and how their performances can be improved simultaneously.

Generative Adversarial Networks (GANs) (Goodfellow et al., 2014; Gulrajani et al., 2017) are good attempts to couple generative and discriminative models in an adversarial manner, where a two-player minimax game between learners is carefully crafted. GANs have achieved tremendous success in domains such as image generation (Denton et al., 2015), and related studies have also shown their effectiveness in semi-supervised learning (Salimans et al., 2016; Kumar et al., 2017). However, GANs are notoriously difficult to train, most training objectives work well for only one model, either the discriminator or the generator, so rarely both learners can be optimal at the same time (Arjovsky & Bottou, 2017; Wiatrak et al., 2019). This essentially arises from the adversarial nature of GANs,

that during the process, optimizing one learner can easily destroy the learning ability of the other, making GANs fail to converge.

Another limitation of simultaneously optimizing the generator and the discriminator comes from the discrete nature of text in NLP, as no gradient propagation can be done from discriminators to generators. One theoretically sound attempt is to use reinforcement learning (RL), but the sparsity and the high variance of the rewards in NLP make the training particularly unstable (Caccia et al., 2020).

To address these shortcomings, we novelly introduce a self-consistent learning framework based on one generator and one discriminator: the generator and the discriminator are alternately trained by way of cooperation instead of competition, and the samples are used as the medium to pass the feedback signal from the discriminator. Specifically, in each round of training, the samples generated by the generator are synthetically labeled by the discriminator, and then only part of them would be selected based on dynamic thresholds and used for the training of the discriminator and the generator in the next round. Several benefits can be discovered from this cooperative training process. First, a closed-loop form of cooperation can be established so that we can get the optimal generator and discriminator at the same time. Second, this framework helps improve the generation quality while ensuring the domain specificity of generator, which in turn contributes to training. Third, a steady stream of diverse synthetic samples can be added to the training in each round and lead to continuous improvement of the performance of all learners. Finally, we can start the training with only domain-related corpus and obtain strong results, while these data can be easily sampled with little cost or supervision. Also, the performance on labeled datasets can be further boosted based on the SOTA level. As an example to demonstrate the effectiveness of our framework, we examine it on the task of sentence semantic matching. The experiments show that our method significantly improves over standalone state-of-the-art discriminative models on zero-shot and full-data settings.

Our contributions are summarized as follows,

• We propose a self-consistent learning framework that incorporates the generator and the discriminator, in which both achieve remarkable performance gains simultaneously.

• We propose a dynamic selection mechanism such that cooperation between the generator and the discriminator drives the convergence to reach their scoring consensus.

• Experimental results show that our proposed framework significantly outperforms the state-of-the-art methods for the task of sentence semantic matching.

## 2 RELATED WORKS

To alleviate the lack of annotated data in supervised learning in NLP, semi-supervised learning (SSL) has been a popular line of research (Van Engelen & Hoos, 2020). The sources of the unlabeled data required by SSL are either collected from the domains or generated by generative language models. Then NLU models can learn from the unlabeled data by pseudo-labeling (Arazo et al., 2020; Banitalebi-Dehkordi & Zhang, 2021) and consistent regularization (Jeong et al., 2019; Sohn et al., 2020). However, collecting unlabeled data comes at a cost(though smaller than labeling data), and the total amount is limited. Even with generative models, there is no guarantee of the quality of the generated samples, because the model cannot tune the generating results based on the performance of the downstream tasks. In contrast, our method usually includes a continuously updated generative model, which dynamically adjusts its generation according to the performance of downstream tasks.

In GANs, the generator is adversarially trained with the discriminator. Unlike conventional GANs in continuous domains, language GANs usually employ Gumbel-Softmax differentiation (Jang et al., 2017; Yin et al., 2020), Reinforcement Learning (RL) (Yu et al., 2017; Wu et al., 2021), or modified training objectives (Montahaei et al., 2021) to update the generator, to use the non-differential signals from the discriminator. However, language GANs are often criticized for underperforming Maximum likelihood estimation (MLE) and are very difficult to train, even the single optimality of either the generator or the discriminator cannot be guaranteed (Alvarez-Melis et al., 2022). In comparison, our proposed framework allows us to cooperatively couple the generator and the discriminator, leading to continuous improvement for both learners.

## 3 METHODOLOGY

### 3.1 COOPERATIVE OR ADVERSARIAL

Following the principle of self-consistency outlined in Ma et al. (2022), a closed-loop training needs to be built between the generator and the discriminator, either cooperatively or adversarially. GANs are typical examples of adversarial learning, but training GANs remains quite unstable. Let us consider an extreme case to show the possible instability: the discriminator can perfectly distinguish real data and fake data generated by the generator, and the generator can fully reproduce the real data distribution. Then the discriminator has only a 50% probability of selecting all samples that are generated by the generator. Therefore, any further updates to the generator parameters based on the feedback from the discriminator deviate the generator from the optimum. Neither the generator nor the discriminator can likely be optimal (Arjovsky & Bottou, 2017; Lamprier et al., 2022). In practice, a very delicate balance needs to be maintained between the discriminator and the generator to keep the training stable. In terms of cooperatively closed-loop learning, as discussed below, it does not suffer from instability: the generator and the discriminator usually enhance each other.

### 3.2 SELF-CONSISTENT LEARNING FRAMEWORK

In this section, we introduce our self-consistent learning (**SCL**) framework.

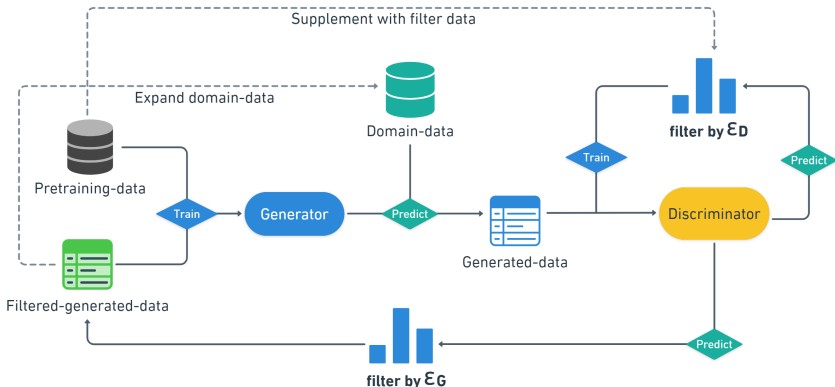

Figure 1: Overview of the flow chart for the SCL framework.

As shown in Figure 1, our framework, similar to the GANs, consists of a generator and a discriminator model. However, contrasting to the GANs, these two parts in our framework work cooperatively to enhance each other. Specifically, for any given class $k$, the generator $\mathcal{G}$ now become a conditional generator that takes in an input sentence $s_k^a$ and generate an output sentence $s_k^b$. The discriminator $\mathcal{D}$ is then responsible for discriminating the sentence using a dynamic threshold $\epsilon_{\mathcal{D}}$. The discriminated sentence is used as positive or negative data for that specific class to continue the training process. Once the new discriminator is trained, the sentence is discriminated again by the new discriminator with a different dynamic threshold $\epsilon_{\mathcal{G}}$. This time only the positive data is passed to the generator as the training data for the new round. In this way, a closed loop of cooperation is formed.

In the above closed-loop training, we propose a **selection mechanism** that uses dynamic thresholds to filter samples. This mechanism is empirically shown to play a critical role in closing the gap between the generator and the discriminator, and thus makes this cooperation loop a virtuous circle. Specifically, as shown in Equation 1, the output probability $p_{\mathcal{D}}(y = k|s_k^b)$ that the sentence $\{s_k^b\}$ belongs to class $k$ is calculated from the embedding representation $\mathbf{H}^1$ of $\{s_k^b\}$,

$$p_{\mathcal{D}}(y = k|s_k^b) = \text{softmax}(\text{MLP}(\mathbf{H})) \tag{1}$$

---

[1]We follow Reimers & Gurevych (2019) and use the embedding representation of $CLS$-token as the sentence representation $\mathbf{H}$.

where $y$ represents the class label. Then, through the filtering function $\texttt{filter}_k^{(t)}(\cdot)$ in round $t$ for the $k$-th class in Equation 2, we keep samples whose output probability is not less than threshold $\epsilon_{t,k}$, while other generated samples whose confidence is lower than threshold $\epsilon_{t,k}$ are discarded.

$$\texttt{filter}_k^{(t)}(s_k^b) \triangleq p_{\mathcal{D}}(k|s_k^b) \geq \epsilon_{t,k} \tag{2}$$

where $\epsilon_{t,k}$ represents the dynamic threshold for accepting $\{s_k^b\}$ as negative or positive samples in the $t$-th round. The generalized threshold function for $\epsilon_{t,k}$ is defined as,

$$\epsilon_{t,k} = f(t, \mathcal{L}_{t-1,k}, \epsilon_{t-1,k}) \tag{3}$$

where $\mathcal{L}_{t-1,k}$ and $\epsilon_{t-1,k}$ represent the discriminator loss and threshold for round $t-1$, respectively. $\mathcal{L}_{0,k}$ is set as 0 and $\epsilon_{0,k} = \lambda$, where $\lambda$ represents a hyperparameter.

**Theorem 1** *At round $t$, given the previous round discriminator $\mathcal{D}_\phi^{t-1}$, the aim of optimizing the generator $\mathcal{G}_\theta^t$ boils down to:*

$$\min_\theta \mathbb{D}_{KL}(p_{\mathcal{D}_\phi^{t-1}}^k(\cdot), p_{\mathcal{G}_\theta^t}^k(\cdot))$$

*where $\mathbb{D}_{KL}$ is the standard KL divergence, $p_{\mathcal{G}_\theta^t}^k(\cdot)$ refers to the degree of confidence that the sentences generated by the generator belongs to a given class $k$ (we can either train the generator to express its confidence in the generated sentences(Lin et al., 2022) or use a fixed third-party model to score them(Gao et al., 2021)), and $p_{\mathcal{D}_\phi^{t-1}}^k(\cdot)$ the probability of being classified into class $k$ given by the discriminator.*

Theorem 1 shows that the generator at round $t$ is encouraged to approximate the probability distribution given by the previous round discriminator. The proof is given in Appendix A. In particular, on the basis of a well-pretrained discriminator, the generated distribution of the generator can be guaranteed to be faithful to the real data distribution.

**Why Cooperative, Not Adversarial?** (1) the generator is no longer a challenger to the discriminator that only provides negative data points to fool it, but now serves as a data augmenter to provide both positive and negative data points to enhance the discriminator; (2) the generator no longer updates its parameters through the policy gradients guided by the signals from the discriminator, but rather by utilizing the filtered data points to further improve its conditional generation quality. Note that by deliberately choosing the conditional generation paradigm along with the selection mechanism, we not only make the training more stable due to the different training goals, but also bypass the mode collapse problem of GANs (see Section 4.6 for further discussion). Besides, by iterating through the loops, our framework achieves self-consistency by honing the domain specificity of the generator and increasing the domain data exposure of the discriminator.

### 3.3 SENTENCE SEMANTIC MATCHING

We leverage the sentence semantic matching task (*i.e.* $k = 2$) as an example to demonstrate the effectiveness of our method. At this time, corresponding to Equation 2, $k = 1/0$ represents the positive/negative class, and $\texttt{filter}_{1/0}^{(t)}$ represents the filter function in round $t$ for the positive/negative class respectively. First, let us introduce the formal definition of this task. Given two sentences $s^a = \{w_1^a, w_2^a, ..., w_{\ell_a}^a\}$ and $s^b = \{w_1^b, w_2^b, ..., w_{\ell_b}^b\}$, where $w_i^a$ and $w_j^b$ represent the $i$-th and $j$-th tokens in the sentences, and $\ell_a$ and $\ell_b$ indicate the length of $s^a$ and $s^b$. The goal of this task is to learn a discriminator $\mathcal{D}$ to precisely predict the label $y = \mathcal{D}(s^a, s^b)$, where $y \in \mathcal{Y} = \{0, 1\}$ indicates whether the two sentences are similar.

In our task, $\mathcal{G}$ is trained to generate a similar sentence $s^b$ from any given sentence $s^a$ and $\mathcal{D}$ is trained to predict label $y$ from any given sentence pair $\{s^a, s^b\}$. As demonstrated in Figure 1, there are mainly two training processes in the entire framework: fix $\mathcal{G}$ to train $\mathcal{D}$ and fix $\mathcal{D}$ to train $\mathcal{G}$. We introduce the two training procedures in detail with the $t$-th round training.

**Training** $\mathcal{D}$: We first randomly sample $s_t^a$ from domain-related corpus $C$, and then input $s_t^a$ to $\mathcal{G}^t$ to generate $s_t^b$. Next, we feed sentence pair $\{s_t^a, s_t^b\}$ into $\mathcal{D}^{t-1}$ to predict the label $y_{t-1}$, and filter $\{s_t^a, s_t^b, y_{t-1}\}$ using threshold $\epsilon_\mathcal{D}^{t-1}$. Finally, we train $\mathcal{D}^{t-1}$ on the selected data and pre-training data $P$ to get an improved discriminator $\mathcal{D}^t$. Note that the filtered data have both positive and negative samples. The update process of $\mathcal{D}$ seeks to minimize the cross-entropy loss over all instances:

$$\mathcal{L}_\mathcal{D}(\boldsymbol{s}, \boldsymbol{y}) = \frac{1}{|\boldsymbol{s}|} \sum_{i=1}^{|\boldsymbol{s}|} -[y_i \cdot \log p_\mathcal{D}(y_i = 1 | s_i^a, s_i^b) + (1 - y_i) \cdot \log(1 - p_\mathcal{D}(y_i = 1 | s_i^a, s_i^b))] \quad (4)$$

**Training** $\mathcal{G}$: We feed the generated sentence pairs $\{s_t^a, s_t^b\}$ into $\mathcal{D}^t$ to predict new labels $y_t$, and then filter $\{s_t^a, s_t^b, y_t\}$ using threshold $\epsilon_\mathcal{G}^t$ and additional rules [2]. Note that the filtered data has only positive samples. For the filtered data, we supplement it with the pre-training data $P$ to update $\mathcal{G}^t$ to $\mathcal{G}^{t+1}$ [3] We also take out $s_t^b$ from the filtered data and add them to the domain-related corpus. The expanded domain corpus are used to sample conditional sentences in the next round of generation. The update procedure of $\mathcal{G}$ employs the negative log-likelihood function over all instances:

$$\mathcal{L}_\mathcal{G}(\boldsymbol{s^a}, \boldsymbol{s^b}) = -\frac{1}{|\boldsymbol{s^b}|} \sum_{t=1}^{|\boldsymbol{s^b}|} \log p_\mathcal{G}(s_t^b | s_{<t}^b, \boldsymbol{s^a})$$

For the selection mechanism, we adopt the form $\epsilon^t = m * t + \lambda$ after comparing the effects of different functions through experiments according to Equation 3. where $m$ is the increment of the threshold for each round, $\lambda$ is the initial threshold, and $\epsilon_t$ is the threshold for rounds $t$.

In the process of training $\mathcal{G}$, since the sentences generated in each round are added to the domain-related corpus, the source of domain-specific data is thus monotonically expanding by iterating the self-consistent learning loop. The formalized process is shown in Algorithm 1.

---

**Algorithm 1** Self-consistent Learning (**SCL**)

---

**Require:** Generator $\mathcal{G}$; Discriminator $\mathcal{D}$; Domain-Related Corpus $C$; Pre-training Data $P$.
  1: Initialize $\mathcal{G}^0$ and $\mathcal{D}^0$ with pre-trained language models;
  2: Warm-up $\mathcal{G}^0$ and $\mathcal{D}^0$ with pre-training data $P$ to get $\mathcal{G}^1$ and $\mathcal{D}^1$;
  3: **for** each round $i \in [1, n]$ **do**
  4:    **if** Two consecutive rounds of discriminator still improve **then**
  5:       Generate similar sentences $s^b \sim p_{\mathcal{G}^i}(\cdot | s^a)$ from sampled sentences $s^a$ from $C$;
  6:       Predict pseudo-labels $y^i \sim p_{\mathcal{D}^i}(\cdot | s^a, s^b)$;
  7:       Use threshold $\epsilon_\mathcal{D}^i$ to select data on $\{s^a, s^b, y^i\}$ to train $\mathcal{D}^{i+1}$;
  8:       Predict pseudo-labels $y^{i+1} \sim p_{\mathcal{D}^{i+1}}(\cdot | s^a, s^b)$;
  9:       Use threshold $\epsilon_\mathcal{G}^i$ and additional rules to select data on $\{s^a, s^b, y^{i+1}\}$ to train $\mathcal{G}^{i+1}$;
 10:    **end if**
 11: **end for**

---

## 4 EXPERIMENTS

### 4.1 TASKS DESIGN

In our experiments, the pre-training datasets are used to warm up the discriminator and generator, and the domain-related corpus is a set of independent sentences. To avoid label leakage, none of the

---

[2]The additional rules are used to exclude sentences which are too long, too short, or too similar according to the longest common substring algorithm.

[3]Note that the pre-training data $P$ is used to warm up $\mathcal{G}$ and $\mathcal{D}$. Although pre-training data is not mandatory in subsequent training, we empirically found that including it when training $\mathcal{G}$ can prevent language degeneration and improve downstream performances.

training datasets participate in the pre-training of the generator and discriminator. In other words, the datasets in pre-training and self-consistent training are two non-overlapped datasets.

**Zero-Shot Baseline**: We utilize the best-performing Chinese model RoBERTa-wwm-ext-large (Cui et al., 2020; 2021) and English model ALBERT-xxlarge-v2 (Lan et al., 2020) as the base discriminators in our self-consistent learning framework.

**Fine-Tune Baseline**: We compare our model with several state-of-the-art semantic matching models, including the following. Chinese models MacBERT (Cui et al., 2020), StructBERT (Wang et al., 2020), RoFormer (Su et al., 2021; Su et al., 2022), XLNet (Xu et al., 2020), ELECTRA (Cui et al., 2020), ALBERT (Lan et al., 2020), RoBERTa (Cui et al., 2020; 2021) and English models BERT (Devlin et al., 2019b), XLM-RoBERTa (XLM-R) (Conneau et al., 2020), XLNet (Yang et al., 2019b), ELECTRA (Clark et al., 2020), ALBERT (Lan et al., 2020), RoBERTa (Liu et al., 2019). For a fair comparison, we use models that are as close in size as possible.

## 4.2 EXPERIMENTS SETUP

### 4.2.1 DATASETS

We conduct experiments on three Chinese semantic matching datasets AFQMC (Xu et al., 2020), CHIP-STS (Zhang et al., 2022a), Chinese-QQP (Wang et al., 2019) and an English semantic matching dataset MRPC (Wang et al., 2019). More details about the datasets are given in Appendix E.

### 4.2.2 MODEL PRE-TRAINING

We adopt the well-established Transformer-XL (Dai et al., 2019)/OPT (Zhang et al., 2022b) architectures as the Chinese/English generator. To enable the generator to generate similar sentences with better linguistic quality, we pre-train a Transformer-XL model with 5.0 billion parameters and incrementally pre-train an OPT model with 2.7 billion parameters on the corpus consisting of plain texts and similar sentence pairs. Cleaned large-scale Chinese corpus WuDaoCorpora (Yuan et al., 2021) and English corpus WikiText (Merity et al., 2017) are used as plain texts. Similar sentence pairs that do not overlap with downstream datasets are used in the pre-training, and the designed prompts are employed to guide the generation of similar sentences. More details regarding model pre-training can be found in Appendix D.

## 4.3 ZERO-SHOT RESULTS

Table 1 shows how the F1 score of the discriminator varies with the number of self-consistent learning rounds on different datasets in the zero-shot task. According to Algorithm 1, the training is stopped when the discriminator no longer improves for two consecutive rounds. In addition, these four datasets are collected from different domains to further reflect the generality of our method in different domains. Specific training settings are recorded in Appendix F.

Table 1: Zero-Shot Performance of the Discriminator. (F1 score (%))

| Round Num | AFQMC (*Financial*) | CHIP-STS (*Medical*) | Chinese-QQP (*Common*) | MRPC (*News*) |
|---|---|---|---|---|
| baseline | 38.25 | 58.82 | 57.88 | 68.54 |
| 1 | 39.61 | 62.89 | 60.08 | 75.47 |
| 2 | 44.98 | 67.24 | 58.57 | 76.63 |
| 3 | 45.99 | 71.38 | 60.30 | 83.00 |
| 4 | 45.71 | 71.45 | 61.31 | 83.90 |
| 5 | 48.01 | 74.06 | 64.47 | 84.24 |
| 6 | 50.41 | 74.08 | 66.44 | 84.50 |
| 7 | 50.68 | 76.66 | 63.88 | 84.32 |
| 8 | **51.36** | 76.30 | 65.46 | **84.61** |
| 9 | - | 76.67 | 68.08 | - |
| 10 | - | **77.42** | **70.51** | - |
| | **+13.11** | **+18.60** | **+12.63** | **+16.07** |

The scores in the last line of Table 1 give the improvement of our discriminator in the last round relative to the first round. We can see that the F1 score gradually increases after each training round, eventually reaching a 10+ absolute percentage (AP) improvement. We believe what drives the improvement of the discriminator is the self-consistency, which it acquires with the generator step by step during the loop.

To verify that the generator also improves after self-consistent training, we adopt Perplexity and Bertscore (Zhang et al., 2020) to measure the language fluency and the semantic similarity (i.e. domain specificity) respectively. For different generators in different rounds, we first select $s^a$ in similar sentence pairs from the same test set as the original sentences input, and generate similar sentences $s^b$ with greedy search. The reason for not using other sampling methods is to ensure reproducibility. Given the generated sentences, we introduce an additional GPT2 [4] model to calculate the perplexity of generated similar sentences, and use a third-party library [5] to calculate the bertscore between the original and generated similar sentences. The results are shown in Table 2.

Table 2: Zero-Shot Performance of the Generator.

|  | AFQMC | CHIP-STS | Chinese-QQP | MRPC |
|---|---|---|---|---|
| Perplexity -first round | 10.13 | 6.86 | 12.94 | 28.71 |
| Perplexity -last round | 8.43 | 5.97 | 12.27 | 17.56 |
| Bertscore -first round | 0.79 | 0.84 | 0.87 | 0.94 |
| Bertscore -last round | 0.80 | 0.85 | 0.89 | 0.97 |

We can see that the perplexity/bertscore of the last round in Table 2 has decreased/improved compared to the first round. Note that a lower perplexity indicates a more fluent sentence, while a higher bertscore indicates a more similar sentence. It suggests that after self-consistent training, the generator is gradually improved in language fluency and semantic similarity (i.e. domain specificity). The reason why the improvement of the generator is not as obvious as that of the discriminator is that the size of the generator is several times that of the discriminator, and the total number of training samples is limited. In Appendix G, the generated samples of the generator in different rounds are given to show the changes in the generation.

## 4.4 FINE-TUNE RESULTS

Our method not only works well in the zero-shot case, but also achieves good results in the full-data case. For the sake of a fair comparison, we reproduce several state-of-the-art semantic matching models on the four training sets, and their performances on the test sets are shown in Table 3.

Our approach uses the best-performing model on a single test set as the base discriminator for self-consistent learning. The bold scores in the last line of Table 3 show that our method outperforms the SOTA results (shaded in gray) by 1 to 2 AP on all four test datasets, indicating the potential of self-consistent learning to further improve the model performance and establish new SOTA.

## 4.5 EVALUATING SELF-CONSISTENCY

In this section, we evaluate the consistency between the generator and the discriminator as the learning loop unfolds. We follow the same method used in Section 4.3 and use greedy search to generate similar sentences on the same test set. Then we take the confidence of the discriminator $R_{\mathcal{D}}$ as the score of the discriminator, which is calculated for the original sentences $s^a$ and the generated similar sentences $s^b$ according to Equation 5.

---

[4]Wenzhong-GPT2-110M (Wang et al., 2022) for Chinese data, and GPT2-base (Radford et al., 2019) for English data.

[5]https://pypi.org/project/bert-score/

Table 3: F1 Score(%) of Different Discriminators on the Test Datasets.

| | #Param(zh/en) | AFQMC | CHIP-STS | Chinese-QQP | MRPC | AVG |
|---|---|---|---|---|---|---|
| BERT$_{large}$ | -/335M | - | - | - | 82.51 | - |
| XLM-R$_{base}$ | -/278M | - | - | - | 84.27 | - |
| MacBERT$_{large}$ | 326M/- | 61.11 | 85.94 | 72.94 | - | 73.33 |
| StructBERT$_{large}$ | 326M/- | 60.56 | 85.17 | 76.33 | - | 74.02 |
| RoFormer$_{large}$ | 316M/- | 64.19 | 84.16 | 76.56 | - | 74.97 |
| XLNet$_{large}$ | 360M/360M | 50.31 | 82.97 | 64.96 | 79.51 | 69.44 |
| ELECTRA$_{large}$ | 324M/334M | 54.59 | 84.97 | 71.81 | 89.64 | 75.25 |
| ALBERT$_{large}$ | 221M/223M | 56.87 | 86.32 | 70.52 | 91.21 | 76.23 |
| RoBERTa$_{large}$ | 326M/355M | 57.29 | 86.93 | 74.58 | 90.24 | 77.26 |
| Our Method | - | **66.59** | **88.39** | **78.43** | **92.78** | **81.55** |

$$R_{\mathcal{D}} = p_{\mathcal{D}}(y^+ | s^a, s^b) \tag{5}$$

where $y^+$ represents a positive label.

However, for the generator, to the best of our knowledge, there is no reliable way to measure how similar $s^a$ and $s^b$ are by using the generator itself. Therefore, to quantify this similarity, we introduce a third-party static model SimCSE [6] to get the embedding representation $\mathbf{A}, \mathbf{B}$ of sentences $s^a, s^b$. The cosine similarity $R_{\mathcal{G}}$ between $\mathbf{A}$ and $\mathbf{B}$ is then calculated according to Equation 6 to approximate the score of the generator.

$$\mathbf{A}, \mathbf{B} = \text{Encoder}(s^a), \text{Encoder}(s^b)$$
$$R_{\mathcal{G}} = \frac{\mathbf{A} \cdot \mathbf{B}}{\|\mathbf{A}\|_2 * \|\mathbf{B}\|_2} \tag{6}$$

where $\mathbf{A}$ and $\mathbf{B}$ both represent the embedding representation at the $[CLS]$ position. Note that the original sentence $s^a$ remains unchanged in each round, while the generated sentence $s^b$ changes.

Finally, for the trained discriminator and generator in each round $t$, we can obtain two score distributions $\mathbf{R}_{\mathcal{D}}^{\mathbf{t}}$ and $\mathbf{R}_{\mathcal{G}}^{\mathbf{t}}$ correspondingly. According to Theorem 1, we draw the curves of KL divergence between $\mathbf{R}_{\mathcal{D}}^{\mathbf{t}}$ and $\mathbf{R}_{\mathcal{G}}^{\mathbf{t}}$ in each round for the four datasets: AFQMC, CHIP-STS, Chinese-QQP, and MRPC. As illustrated in Figure 2, all the curves show a clear downward trend, indicating that the distance between the two score distributions decreases with the increase in the number of training rounds until a score consensus is reached. Table 4 shows the values of KL divergence in the first and last rounds. Numerically, it is more evident that the distances are significantly reduced on the four datasets.

## 4.6 Effect of Pre-training Data and Selection Mechanism

We perform ablation experiments on the pre-training data and the selection mechanism in the zero-shot case. As described in Section 4.1, the pre-training data is used to pre-train the generator and discriminator, completely independent of the experimental datasets in self-consistent training.

To explore the influence of pre-training data on self-consistent training, we no longer add it in each round when training the discriminator, and only the generated data is used. But when the generator is trained, pre-training data is still retained to prevent language degeneration and lack of expressive diversity of the generation. The result of removing pre-training data is shown as the green curves in Figure 3. With all other training parameters being the same, after the same number of training rounds, the discriminator is slightly worse compared to the original method (red curves in Figure 3).

---

[6]We use SimCSE-BERT-base to calculate scores on Chinese datasets and sup-SimCSE-BERT-base-uncased on English datasets. (Gao et al., 2021)

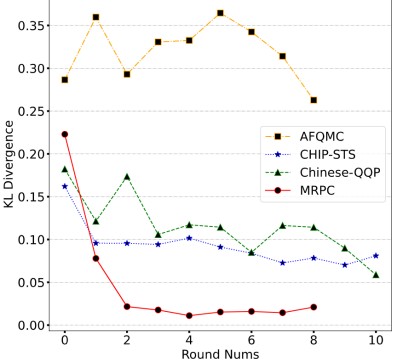

Figure 2: The KL Divergence between the score distributions of the Discriminator and the Generator.

|  | KL Divergence -first round | KL Divergence -last round |
|---|---|---|
| AFQMC | 0.29 | 0.26 |
| CHIP-STS | 0.16 | 0.08 |
| Chinese-QQP | 0.18 | 0.06 |
| MRPC | 0.22 | 0.02 |

Table 4: The KL divergence in the first and last rounds.

However, the green curves maintain an upward trend and are very close to the red curves in all datasets except CHIP-STS. This shows that the generated data plays a key role in continuously improving the discriminator, while the pre-training data has a limited role.

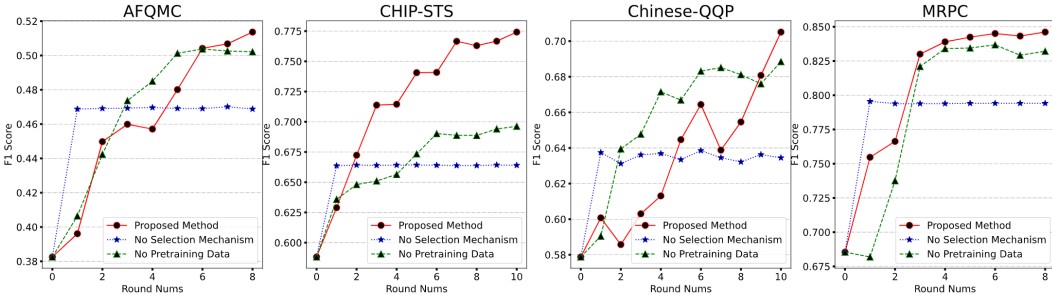

Figure 3: Results of ablation experiments on pre-training data and selection mechanism. Results of the proposed method, results without pre-training data, and results without the selection mechanism are given in red, green, and blue, respectively.

In order to explore the effect of the selection mechanism on training the discriminator, we remove the selection mechanism when training the discriminator, while the training of the generator remains unchanged. The blue curves in Figure 3 depict the performance of the discriminator in each round after removing the selection mechanism. Compared to the original method (red curves), the discriminator only improves in the first round after removing the selection mechanism, which demonstrates the importance of the selection mechanism on the discriminator for the convergence of the self-consistent learning framework.

## 5 CONCLUSION

In this paper, we propose a self-consistent learning framework to enable cooperative training of the generator and the discriminator. During the training process, the generator and the discriminator continuously enhance each other until reaching a score consensus. This framework can utilize both limited labeled data and large-scale unlabeled domain-related corpus. Experimental results on four Chinese/English sentence semantic matching datasets demonstrate that as a form of closed-loop training, our proposed framework can achieve new state-of-the-art results with continuously improved generators and discriminators.

For future work, we will explore the effectiveness of our self-consistent learning framework on more NLP tasks, since the framework is straightforward and has no additional requirements on generators and discriminators.

## 6 REPRODUCIBILITY STATEMENT

We now discuss the efforts that have been made to ensure the reproducibility of our work. We have packaged the executable code and data into supplementary materials, which can be downloaded and run directly. In addition, we also provide detailed experimental parameters in the appendix to reproduce the experimental results. All datasets and code platforms (Pytorch) we use are public.

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

# A  PROOFS

## A.1  PROOF OF THEOREM 1

**Theorem 1** *At round $t$, given the previous round discriminator $\mathcal{D}_\phi^{t-1}$, the aim of the optimization of the generator $\mathcal{G}_\theta^t$, boils down to,*

$$\min_\theta \mathbb{D}_{KL}(p_{\mathcal{D}_\phi^{t-1}}^k(\cdot), p_{\mathcal{G}_\theta^t}^k(\cdot))$$

*where $\mathbb{D}_{KL}$ is the standard KL divergence, $p_{\mathcal{G}_\theta^t}^k(\cdot)$ refers to the degree of confidence that the sentences generated by the generator belong to a given class $k$ (we can either train the generator to express its confidence in the generated sentences(Lin et al., 2022) or use a fixed third-party model to score them(Gao et al., 2021)), and $p_{\mathcal{D}_\phi^{t-1}}^k(\cdot)$ the probability of being classified into class $k$ given by the discriminator.*

**Proof.** We use the previous round generator $\mathcal{G}_\theta^{t-1}$ to generate samples, and filter them using the previous round discriminator $\mathcal{D}_\phi^{t-1}$ with a threshold $\epsilon_{t-1}$, then these samples are used for training of the current round generator $\mathcal{G}_\theta^t$. Therefore, the optimization of $\mathcal{G}_\theta^t$ will tend to maximize the probability that the generated samples pass the discrimination for the fixed $\mathcal{D}_\phi^{t-1}$. For a given class $k$, we have

$$\max_\theta \mathbb{E}_{x \sim p_{\mathcal{G}_\theta^{t-1}}^k} p_{\mathcal{G}_\theta^t}^k(x) \quad \texttt{s.t.} \quad \texttt{filter}_k^{(t-1)}(x) = 1$$

where the definition of function $\texttt{filter}_k^{(t-1)}(\cdot)$ has been given in Equation 2.

The above objective is equivalent to sampling from the generator being optimized in round $t$ and making these samples pass the discrimination in round $t-1$ as much as possible, which gives

$$\max_\theta \mathbb{E}_{x \sim p_{\mathcal{G}_\theta^t}^k} p_{\mathcal{D}_\phi^{t-1}}^k(x)$$

where $p_{\mathcal{D}_\phi^{t-1}}^k(x)$ is fixed.

A further transformation of the formula shows that

$$\max_\theta \mathbb{E}_{x \sim p_{\mathcal{G}_\theta^t}^k} p_{\mathcal{D}_\phi^{t-1}}^k(x)$$
$$\stackrel{(i)}{\Rightarrow} \max_\theta \int \mathrm{d}\boldsymbol{\theta} \nabla_{\boldsymbol{\theta}} \mathbb{E}_{x \sim p_{\mathcal{G}_\theta^t}^k} p_{\mathcal{D}_\phi^{t-1}}^k(x)$$
$$\stackrel{(ii)}{\Rightarrow} \max_\theta \int \mathrm{d}\boldsymbol{\theta} \mathbb{E}_{x \sim p_{\mathcal{G}_\theta^t}^k} \nabla_{\boldsymbol{\theta}} \log p_{\mathcal{G}_\theta^t}^k(x) p_{\mathcal{D}_\phi^{t-1}}^k(x)$$
$$\stackrel{(iii)}{\Rightarrow} \max_\theta \int \mathrm{d}\boldsymbol{\theta} \nabla_{\boldsymbol{\theta}} \frac{1}{N} \sum_{i=1}^N \{\log p_{\mathcal{G}_\theta^t}^k(x_i) p_{\mathcal{D}_\phi^{t-1}}^k(x_i) - \log p_{\mathcal{D}_\phi^{t-1}}^k(x_i) p_{\mathcal{D}_\phi^{t-1}}^k(x_i)\}$$
$$\stackrel{(iv)}{\Rightarrow} \min_\theta \mathbb{D}_{KL}(p_{\mathcal{D}_\phi^{t-1}}^k(\cdot), p_{\mathcal{G}_\theta^t}^k(\cdot))$$

where $(i)$ uses the integral property that integrating the derivative of a function gives the original function along with a constant, $(ii)$ takes advantage of the derivative property of the logarithmic function, $(iii)$ approximates the expectation of the probability distribution $p_{\mathcal{G}_\theta^t}^k(\cdot)$ by using averaging on $N$ samples sampling from $p_{\mathcal{G}_\theta^t}^k(\cdot)$, and adding a constant term with respect to $\theta$ under the summation would not change its derivative, and $(iv)$ cancels out the integral and the derivative and uses the definition of KL divergence. The above concludes our proof.

# B  EXPERIMENTS ON DIFFERENT FILTER FUNCTIONS

To compare the effect of different filter functions on the final result, we use four type of functions, including oscillatory function (cosine), constant function and monotonically increasing functions

(quadratic and linear). For the fairness of comparison, we keep the maxima and minima the same for all functions(except for the constant threshold), and the values are given in Appendix F. In addition, the number of training rounds for different functions on the same dataset remains the same.

In the results below, the best results and the second-best results are bold and underlined, respectively.

Table 5: Performance of Different Filter Functions in Zero-Shot Setting. (F1 Score(%))

|  | AFQMC | CHIP-STS | Chinese-QQP | MRPC | AVG |
|---|---|---|---|---|---|
| Baseline | 38.25 | 58.82 | 57.88 | 68.54 | 55.87 |
| Cosine | 47.38 | 74.26 | 64.39 | 83.48 | 67.38 |
| Constant | 47.06 | 74.15 | 68.67 | 84.11 | 68.50 |
| Quadratic | **51.75** | 73.09 | **70.85** | 83.48 | 69.79 |
| Linear | 51.36 | **77.42** | 70.51 | **84.61** | **70.98** |

As can be seen from the Table 5, in the zero-shot setting, the chosen linear function outperforms the other functions, and all the filter functions show an averaging 10+ AP improvement relative to the baseline. Therefore, the self-consistent learning framework makes it easy to choose a certain threshold function and perform well, and the results are not so sensitive to the choice of the functions.

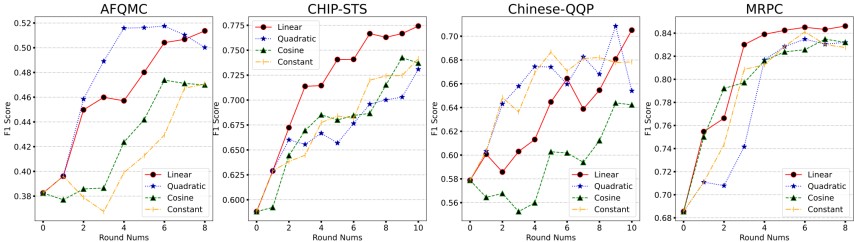

Figure 4: Results of contrast experiments on Cosine(green), Constant(orange), Quadratic(blue) and Linear(red) function.

Figure 4 dipicts the comparison results in each round. The linear function (red line) is significantly better than the other functions on both CHIP-STS and MRPC datasets. In the AFQMC and Chinese-QQP datasets, the quadratic function (blue line) is slightly more effective than the linear function. In general, we can intuitively see that all functions show a significant increase relative to the starting point.

Table 6: Performance of Different Filter Functions in Fine-Tune Setting. (F1 Score(%))

|  | AFQMC | CHIP-STS | Chinese-QQP | MRPC | AVG |
|---|---|---|---|---|---|
| Baseline | 64.19 | 86.93 | 76.56 | 91.21 | 79.72 |
| Cosine | 66.43 | 88.01 | 77.33 | 92.63 | 81.10 |
| Constant | 66.57 | 88.15 | 78.45 | 92.51 | 81.42 |
| Quadratic | 66.37 | 87.76 | **79.26** | 92.75 | 81.54 |
| Linear | **66.59** | **88.39** | 78.43 | **92.78** | **81.55** |

Table 6 shows the effects of different filter functions in the fine-tune experiment. It can be seen that all functions have a $1 \sim 2$ AP increase relative to the baseline, and the chosen linear function achieves the best performance on all datasets except Chinese-QQP.

## C  CONTRASTIVE EXPERIMENTS WITH ADVERSARIAL TRAINING

In this section, We further demonstrate the superiority of the cooperative approach by comparing the results with adversarial experiments. All experimental settings independent of the training method remain the same in the adversarial training.

During the experiments, the generator is no longer trained using the samples filtered by the discriminator, but the rewards passed by the discriminator assist the training. All generated samples are treated as negative samples when training the discriminator.

Specifically, $\mathcal{G}$ takes the prompt ' "$s^a$" is similar to " ' and the first $M$ tokens of $s^b$ as input to get $M$ sentence pairs $< s^a, s^b_m >$, where $m$ is from 1 to $M$. Note that we repeat the process of generating sentences $N$ times to reduce the negative impact caused by the large variance of the rewards.[7] The sentence pair $g^n_m$ of the $m$-th token at the $n$-th time is formalized as

$$g^n_m \leftarrow < s^a, s^b_m > = \mathcal{G}_\theta(s^b_m | s^b_{<m}, \boldsymbol{s^a}; N)$$

Once the $M * N$ sentence pairs $g^n_m$ are generated, they are passed as input to the $\mathcal{D}$ to obtain the probability score $Q^n_m$ for each of them. We take the average of $Q^n_m$ over $N$ as the reward $\bar{Q}_m$ corresponding to the $m$-th token. If the sentence length of $s^b$ is greater than $M$, the rewards of the remaining tokens are all the same as those of the $M$-th token. Taking the $m$-th token as an example, the rewards $\bar{Q}_m$ can be formalized as

$$\bar{Q}^{\mathcal{G}_\theta}_{\mathcal{D}_\phi}(m) = \left\{ \begin{array}{cc} \frac{1}{N} \sum_{n=1}^N \mathcal{D}_\phi(g^n_m) & m \leq M \\ \bar{Q}(M) & m > M \end{array} \right.$$

Therefore, the objective function for training the generator $\mathcal{G}$ is,

$$\mathcal{L}_{\mathcal{G}}(\boldsymbol{s^a}, \boldsymbol{s^b}) = -\frac{1}{|\boldsymbol{s^b}|} \sum_{t=1}^{|\boldsymbol{s^b}|} \log(p_{\mathcal{G}}(s^b_t | s^b_{<t}, \boldsymbol{s^a}) * \bar{Q}_t)$$

The loss function of training the discriminator remains the same as Equation 4, but differing from cooperative training, the generated samples are regarded as negative samples to the discriminator, and the training target for the discriminator can be given by

$$\min_\phi -\mathbb{E}_{X \sim p_{\text{data}}}[\log \mathcal{D}_\phi(X)] - \mathbb{E}_{X \sim p_{\mathcal{G}_\theta}}[\log(1 - \mathcal{D}_\phi(X))]$$

The results of zero-shot and fine-tune on the four datasets are shown in Tables 7 and 8.

As can be seen from Table 7, in the zero-shot setting, training in an adversarial manner does not give any improvement over the baseline. Because the initial discriminator in the zero-shot setting is very weak in distinguishing positive and negative samples, it is reasonable to believe that if all generated samples are considered negative samples from the very beginning, it is difficult for the discriminator to know how to distinguish positive samples. As a result, the F1 scores on both AFQMC and CHIP-STS datasets end up being 0, while the scores on the Chinese-QQP and MRPC datasets fluctuate intensively with the number of rounds, which further validates the instability of the adversarial training in the zero-shot setting.

For the fine-tune experiments, Table 8 shows that training in an adversarial manner can slightly improve the performance on the Chinese-QQP and MRPC datasets, but is still worse than the cooperative training. On the AFQMC and CHIP-STS dataset, adversarial training makes it even worse relative to the baseline. It is worth noting that the whole process of adversarial training is so unstable and it is easy to collapse after a few training rounds.

---

[7]In practice, we take $M = 5$, $N = 5$ for ease of calculation.

Table 7: Zero-Shot Performance of Adversarial Trained Discriminator. (F1 score (%))

| Round Num | AFQMC (*Financial*) | CHIP-STS (*Medical*) | Chinese-QQP (*Common*) | MRPC (*News*) |
|---|---|---|---|---|
| baseline | 38.25 | 58.82 | 57.88 | 68.54 |
| 1 | 0.0 | 8.73 | 21.71 | 4.19 |
| 2 | 0.02 | 7.13 | 49.30 | 7.06 |
| 3 | 0.0 | 0.29 | 42.94 | 5.32 |
| 4 | 0.0 | 1.09 | 41.13 | 0.0 |
| 5 | 0.0 | 0.10 | 43.10 | 1.72 |
| 6 | 0.0 | 0.39 | 34.30 | 67.38 |
| 7 | 0.0 | 0.20 | 42.62 | 48.31 |
| 8 | 0.0 | 0.20 | 34.95 | 37.97 |
| 9 | - | 0.20 | 41.81 | - |
| 10 | - | 0.20 | 40.00 | - |

Table 8: Fine-Tune Performance of the Discriminator. (F1 score (%))

| | AFQMC | CHIP-STS | Chinese-QQP | MRPC | AVG |
|---|---|---|---|---|---|
| Baseline | 64.19 | 86.93 | 76.56 | 91.21 | 79.72 |
| Adversarial | 58.37 | 80.46 | 77.93 | 92.18 | 77.24 |
| Cooperative (Our Method) | **66.59** | **88.39** | **78.43** | **92.78** | **81.55** |

## D  MODEL DETAILS

The 5.0B Transformer-XL is pre-trained on 32 A100s with 40G memory for 45 days, the batch size is set to 32*8=256. After running 445k steps, the final validation loss reduces to about 2.4. The 2.7B OPT is incrementally trained on the basis of the open-source model.

During the pre-training of the generator model, we utilize the memory-cache mechanism of Transformer-XL and design a special attention mask to concatenate the multiple input sentences into one sample, to reduce the number of the padding token in a batch and therefore increase the number of effective tokens. To make the generation more robust, we add noise to the original sentences by randomly replacing or discarding tokens with a 5% probability. In addition, the prompts that we use for Chinese generation and English generation are as follows,

- Chinese prompt: "$s^a$"的相似句是"$s^b$" (en: A similar sentence to "$s^a$" is "$s^b$".)
- English prompt: "$s^a$" is similar to "$s^b$"

When training the discriminator, following the usage of special tokens in BERT (Devlin et al., 2019a), we use $[SEP]$ to concatenate two sentences and take the embedding at the $[CLS]$ position to represent the whole sentence to predict the label. Moreover, we utilize the mask method in BERT to randomly mask 15% of the input tokens.

## E  DATASET DETAILS

The statistics of the experimental datasets are reported in Table 9.

Other Chinese datasets (LCQMC (Liu et al., 2018), OPPO, PAWS-X-zh (Yang et al., 2019a), BQ (Chen et al., 2018), CCKS, Chinese-STS-B (Wang et al., 2019)) and English datasets (QQP (Wang et al., 2019), STS-B (Wang et al., 2019), PAWS-X-en (Yang et al., 2019a)) are collected and used as the corpus of similar sentence pairs for pre-training the generator.

The Chinese-QQP dataset contains 9000 pieces of data randomly selected and translated from the English QQP dataset, which is then divided into training set and test set in a ratio of 3:2.

Table 9: Statistics of Experimental Datasets

|  |  | AFQMC | CHIP-STS | Chinese-QQP | MRPC(en) |
|---|---|---|---|---|---|
| Zero-Shot | domain-related corpus | 68668 | 32000 | 10800 | 8,152 |
|  | test dataset | 4316 | 4000 | 3600 | 1725 |
| Fine-Tune | training dataset | 34334 | 16000 | 5400 | 4076 |
|  | test dataset | 4316 | 4000 | 3600 | 1725 |

## F  PARAMETER SETTINGS

The training parameters of zero-shot are shown in Table 10. The three thresholds are used to select positive and negative examples for training the discriminator and positive examples for training the generator, respectively. We adopt cosine annealing learning rate decay strategy during training.

Table 10: Parameter Settings of Zero-Shot.

|  | AFQMC | CHIP-STS | Chinese-QQP | MRPC |
|---|---|---|---|---|
| Maximum Threshold *-negative* | 0.8 | 0.9 | 0.95 | 0.95 |
| Minimum Threshold *-negative* | 0.6 | 0.7 | 0.8 | 0.8 |
| Maximum Threshold *-positive* | 0.8 | 0.9 | 0.95 | 0.95 |
| Minimum Threshold *-positive* | 0.6 | 0.7 | 0.8 | 0.8 |
| Maximum Threshold *-generator* | 0.6 | 0.9 | 0.95 | 0.95 |
| Minimum Threshold *-generator* | 0.6 | 0.7 | 0.8 | 0.8 |
| Threshold Increase | 0.07 | 0.1 | 0.05 | 0.05 |
| Sentence Num | 6000 | 6000 | 3000 | 4000 |
| Learning Rate | | 2e-5 | | |
| Warm Up Steps | | 40 | | |
| Early Stopping Patience | | 1 | | |
| Generator Batch Size *-training* | | 2(concatenate 30 samples) | | 24 |
| Generator Batch Size *-predicting* | | 512 | | 100 |
| Discriminator Batch Size *-training* | | 64 | | 32 |
| Discriminator Batch Size *-predicting* | | 384 | | 96 |

The training parameters of fine-tuning are shown in Table 11.

## G  GENERATE SAMPLES

The generators use nucleus sampling (Holtzman et al., 2020) to generate similar sentences. Generated examples in English are shown in Table 12 and in Chinese in Table 13.

Table 11: Parameter Settings of Fine-Tune.

| | AFQMC | CHIP-STS | Chinese-QQP | MRPC |
|---|---|---|---|---|
| Maximum Threshold *-negative* | 0.98 | 0.98 | 0.84 | 0.8 |
| Minimum Threshold *-negative* | 0.9 | 0.7 | 0.6 | 0.6 |
| Maximum Threshold *-positive* | 0.98 | 0.98 | 0.98 | 0.8 |
| Minimum Threshold *-positive* | 0.9 | 0.7 | 0.9 | 0.6 |
| Maximum Threshold *-generator* | 0.98 | 0.98 | 0.98 | 0.8 |
| Minimum Threshold *-generator* | 0.9 | 0.7 | 0.9 | 0.6 |
| Threshold Increase | 0.07 | 0.07 | 0.07 | 0.2 |
| Sentence Num | 6000 | 6000 | 3000 | 3000 |
| Learning Rate | 5e-6 | | | |
| Warm Up Steps | 40 | | | |
| Early Stopping Patience | 1 | | | |
| Generator Batch Size *-training* | 2(concatenate 30 samples) | | | 24 |
| Generator Batch Size *-predicting* | 512 | | | 100 |
| Discriminator Batch Size *-training* | 32 | 64 | 32 | 32 |
| Discriminator Batch Size *-predicting* | 256 | 384 | 256 | 96 |

Table 12: Generated Samples in English

| MRPC(News) |
|---|
| *origin:* Hong Kong was flat, Australia, Singapore and South Korea lost 0.2-0.4 percent. *first round:* the market is flat, the US is flat, Europe is flat, Japan is flat, ... Malaysia is flat, Philippines is flat, Vietnam is flat, Singapore is *last round:* Hong Kong was unchanged, Australia, Singapore and South Korea were down 0.2-0.4 percent. |
| *origin:* Ballmer has been vocal in the past warning that Linux is a threat to Microsoft. *first round:* Microsoft is a threat to Linux. *last round:* Ballmer has been vocal in the past about the threat Linux poses to Microsoft. |
| *origin:* Last year, Congress passed similar, though less expensive, buyout legislation for peanut farmers, ending that Depression-era program. *first round:* The government is going to buy out your farm and give you a check for $1,000,000.00. You can't refuse. *last round:* Congress passed a similar bill last year to end a Depression-era program for peanut farmers, ending that program. |
| *origin:* Licensing revenue slid 21 percent, however, to $107.6 million. *first round:* The company's net income fell to $1.1 million, or 2 cents per share, in the quarter ended Sept. 30, from $1.3 million, or 3 cents per share, a year earlier. *last round:* Licensing revenue fell 21 percent to $107.6 million in the third quarter. |
| *origin:* SCO says the pricing terms for a license will not be announced for weeks. *first round:* we're not going to tell you how much it costs to buy a car, but we're going to tell you how much it costs to rent one. *last round:* The pricing terms for a license for the SCO software will not be announced for several weeks. |

Table 13: Generated Samples in Chinese.

| | |
|---|---|
| AFQMC (Financial) | *origin*：可以用自己的支付宝为别人的花呗还款吗 
 (Can I use my Alipay to repay someone else's Huabei) 
 *first round*：花呗的钱可以让其他人代还吗 
 (Can someone else pay for the money from Huabei) 
 *last round*：花呗能不能帮别人还 
 (Can my Huabei help others repay?) |
| | *origin*：花呗怎么还欠费 
 (How does Huabei pay off the arrears) 
 *first round*：花呗怎么还有电费 
 (Why does Huabei still have electricity bills) 
 *last round*：花呗还款怎么还 
 (How to pay off Huabei) |
| | *origin*：我的花呗收款二维码 
 (QR code for my Huabei payment) 
 *first round*：商家花呗的二维码 
 (Merchant Huabei's QR code) 
 *last round*：花呗收款二维码在哪里 
 (Where is the QR code for Huabei payment) |
| CHIP-STS (Medical) | *origin*：艾滋病的病因是什么 
 (What is the cause of AIDS) 
 *first round*：艾滋病毒是什么？ 
 (What is HIV?) 
 *last round*：艾滋病是什么原因引起的？ 
 (What causes AIDS?) |
| | *origin*：高血压总是流口水是怎么回事 
 (High blood pressure is always drooling what is going on) 
 *first round*：高血压怎么回事 
 (What about high blood pressure) 
 *last round*：高血压为什么会流口水？ 
 (Why does high blood pressure cause drooling?) |
| | *origin*：得了糖尿病,现在越来越瘦了怎么回事 
 (Why am I getting thinner and thinner now that I have diabetes) 
 *first round*：糖尿病现在怎么回事？ 
 (What's going on with diabetes now?) 
 *last round*：糖尿病患者为什么会瘦？ 
 (Why do people with diabetes lose weight?) |
| Chinese-QQP (Common) | *origin*：如何从此网站删除我的帐户？ 
 (How do I delete my account from this site?) 
 *first round*：怎么删除网站 
 (How to delete a website) 
 *last round*：如何才能删除我的帐户？ 
 (How can I delete my account?) |
| | *origin*：关于电子产品的一些好书是什么？ 
 (What are some good books on electronics?) 
 *first round*：有什么好的电子产品推荐 
 (Any good electronics recommendations) 
 *last round*：有哪些关于电子产品的好书？ 
 (What are some good books about electronics?) |
| | *origin*：为什么没有人看到无尽和无限之间的区别？ 
 (Why does no one see the difference between endless and infinite?) 
 *first round*：为什么宇宙中没有极限的存在 
 (Why is there no limit in the universe) 
 *last round*：为什么没有人知道无限和有限之间的区别？ 
 (Why does no one know the difference between infinite and finite?) |

