# OpenReview forum: "Self-Consistent Learning: Cooperation between Generators and Discriminators"
_ICLR.cc/2023/Conference — Submitted to ICLR 2023_

### Official Review · Reviewer_YrRh · 2022-10-24

**Confidence:** 4
**Correctness:** 2
**Technical Novelty And Significance:** 1
**Empirical Novelty And Significance:** 1
**Recommendation:** 3

**Clarity, Quality, Novelty And Reproducibility:**

The paper is fairly clear. The writing can be significantly improved. The quality of the technical writing especially the proofs are significantly lower than ICLR standards.

The novelty is quite low. Several other works such as https://www.nature.com/articles/s41598-019-52737-x have proposed Data Augmentation schemes with GANs.

The zip file provided with the submission and the code looks that it can be used to reproduce the experiments easily. There are a bunch of differences that haven't been highlighted in the paper which could be improved in the text.

**Details Of Ethics Concerns:**

No Concerns from my side.

**Strength And Weaknesses:**

Strengths:

* Simple technique for assisting the learning of generators and discriminators using a co-operative technique rather than adversarial technique

Weakness:

* Several cooperative GAN based augmentation systems have been proposed and the novelty is extremely low.
* The experiments only show some NLP based experiments and no Vision based experiments.
* Ideally the model should also work with image generation but the conspicous absence indicates the non-generality of the technique
* The proofs are hand wavy and not concrete.

**Summary Of The Paper:**

This paper proposes a co-operative technique for Generator and Discriminator to cooperate and aid each other instead of the standard adversarial game of of the GAN. The proposed technique uses an augmentation scheme to augment the learning data of the generator and data to aid each other's learning progress.

**Summary Of The Review:**

The paper significantly lacks novelty. The proposed technique should ideally also be beneficial to image generation. However, the lack of results raise the suspicion on the generalizability of the technique. The metrics reported on the text based datasets are moderate at best.

---

> ### Author Response · Authors · 2022-11-15
> **Response to Reviewer YrRh**
>
> Thank you for taking the time to review our paper. We provide our responses below.
>
> q1: Several cooperative GAN based augmentation systems have been proposed and the novelty is extremely low.
> > Thank you for recommending the relevant paper[1], please allow us to discuss the fundemental difference for our method from the GAN-based augmentations.
>
> > First, both [1] and [2] conduct experiments in the image domain, but never achieve any positive results in text, which stems from the essential difference between image signals and text sequences, that is, image signals are continuous, while text is discrete (i.e. the sampling process is not differentiable). In fact, according to the results of our adversarial experiments in Appendix C, It can be found that the discriminator cannot converge stably on most datasets when we try to imitate the GAN-based method in text (i.e. through policy gradient).  In contrast, the proposed cooperative training method bypasses the problems of mode collapse and training instability by using samples as feedback signals between generators and discriminators.
>
> > Second, unlike GANs, the goal of our training is not only to obtain a generator that approximates the real distribution for data augmentation, but more importantly,  we are also simultaneously optimizing a discriminator that can be directly used for downstream tasks (e.g. classification tasks). This is guaranteed by the cooperative mechanism of training.
>
> > Finally, for GAN-based methods, the training objective of their discriminators is to distinguish between real samples and generated samples. That is, the discriminator always treats the generated samples as negative samples unconditionally, which makes it impossible for these methods to be used in zero-shot scenarios, where no positive examples exist. (As shown in the zero-shot experiments  in the appendix C). In contrast, in our self-consistent training framework, the training objective of the discriminator is completely the same as the downstream tasks, thus the discriminator shows good performance on zero-shot settings.
>
> q2: The experiments only show some NLP based experiments and no Vision based experiments. q3: Ideally the model should also work with image generation but the conspicuous absence indicates the non-generality of the technique.
> > Currently we are mainly focusing on the textual domain, which is a large enough area to explore. Our intuition is that this framework is applicable to the visual domain, although visual and textual domains are fundamentally different, which may make the transfer difficult. We hope that the future work can use our introduced framework to make further progress in this regard.
>
> q4: The proofs are hand wavy and not concrete.
> > We have added more details to each step of the deraivation in the revised manuscript to make the proof concrete and obvious.
>
> [1] Sandfort V, Yan K, Pickhardt P J, et al. Data augmentation using generative adversarial networks (CycleGAN) to improve generalizability in CT segmentation tasks[J]. Scientific reports, 2019, 9(1): 1-9.
>
> [2]  Zhu J Y, Park T, Isola P, et al. Unpaired image-to-image translation using cycle-consistent adversarial networks[C]//Proceedings of the IEEE international conference on computer vision. 2017: 2223-2232.

---

> > ### Comment · Reviewer_YrRh · 2022-12-09
> > **Satisfied with author's answers**
> >
> > The authors have addressed most of my concerns, I am ready to increase my score to 3. Still it seems not enough contribution for ICLR.

---

### Official Review · Reviewer_8YkB · 2022-10-26

**Confidence:** 2
**Correctness:** 3
**Technical Novelty And Significance:** 2
**Empirical Novelty And Significance:** 3
**Recommendation:** 5

**Clarity, Quality, Novelty And Reproducibility:**

The writing is good, but the proof of the main result needs to be further validated.

**Strength And Weaknesses:**

Strength:
The paper proposes a self-consistent learning framework to enable cooperative training of the generator and the discriminator. The generator and the discriminator are trained alternatively until reaching a score consensus.  This framework makes plenty use of the limited labeled data and large-scale unlabeled domain-related corpus.

Weaknesses:
The paper formulates the main theoretical result into theorem 1, but the proof of theorem in the Appendix is not obvious. In particular, how do you derive from the first line optimization problem to the second optimization problem. Could you provide further detailed proof for the theorem?
The choice of the dynamic threshold looks quite arbitrary, could you provide some criterion for the parameter selection?


**Summary Of The Paper:**

The paper proposes a a self-consistent learning framework to train a discriminator and a generator cooperatively in a closed-loop form. The discriminator and the generator enhance each other during multiple rounds of alternating training. Experimnetal results are given to validate that the proposed framework is easy to train and free from instabilities such as mode collapse and non-convergence.

**Summary Of The Review:**

The paper proposes a self-consistent learning framework to enable cooperative training of the generator and the discriminator. The framework can utilize both limited labeled data and large-scale unlabeled domain-related corpus. But the proof of the main theorem still needs to be validated.

---

> ### Author Response · Authors · 2022-11-15
> **Response to Reviewer 8YkB**
>
> We thank the reviewer for taking the time to review our paper. The suggestions are really helpful in clarifying the narrative and  have been added to the revised manuscript.  We respond to specific questions and concerns below.
>
> q1: The paper formulates the main theoretical result into theorem 1, but the proof of theorem in the Appendix is not obvious. In particular, how do you derive from the first line optimization problem to the second optimization problem. Could you provide further detailed proof for the theorem?
> > The original proof attempts to derive the optimization problem $\min\_{\theta}  \mathbb{D}\_{KL}(p\_{\mathcal{D}^{t-1}\_\phi}^k(\cdot), p\_{\mathcal{G}^{t}\_\theta}^k(\cdot))$ from the problem $\max\_{\theta} \mathbb{E}\_{x\sim  p\_{\mathcal{G}^{t}\_\theta}^k}p\_{\mathcal{D}^{t-1}\_{\phi}}^k(x)$, which shows that in the self-consistent learning, the updating of two adjacent $\mathcal{D}^{t-1}$ model and $\mathcal{G}^{t}$ model can be regarded as optimizing the parameters of $\mathcal{G}^{t}$  to reduce the distribution distance between $p\_{\mathcal{G}^{t}\_\theta}^k(\cdot)$  and $p\_{\mathcal{D}^{t-1}\_{\phi}}^k(\cdot)$. To make this process concrete and obvious, we add more detailed instructions to each step of the derivation in the new manuscript.
>
> q2: The choice of the dynamic threshold looks quite arbitrary, could you provide some criterion for the parameter selection?
> > Experiments on the selection of dynamic thresholds  have been supplemented in the revised manuscript,  please find more details in the Appendix B  (EXPERIMENTS ON DIFFERENT FILTER FUNCTIONS) of the manuscript (main tables are also listed below). Four different threshold functions are used, including cosine, constant, quadratic and linear, and all show improvement compared to the baseline on all experimental settings.
>
> > From the results we can see that monotonically increasing functions(linear and quadratic functions in our case ) are better than other forms of functions( cosine and constant functions), and the linearly increasing function obtains the overall best performance. This indicates that the choice of the dynamic threshold is not tricky at all under the self-consistent learning framework, a linearly increasing function could handle most cases.
>
> Performance of Different Filter Functions in Zero-Shot Setting. (F1 Score(%))
> |           |    AFQMC     |   CHIP-STS   | Chinese-QQP  |     MRPC     |    AVG    |
> | :-------: | :----------: | :----------: | :----------: | :----------: | :-------: |
> | Baseline  |    38.25     |    58.82     |    57.88     |    68.54     |   55.87   |
> |  Cosine   |    47.38     | 74.26 |    64.39     |    83.48     |   67.38   |
> |  Constant|    47.06     |    74.15     |    68.67     | 84.11 |   68.50   |
> | Quadratic |  **51.75**   |    73.09     |  **70.85**   |    83.48     |   69.79   |
> |  Linear   | 51.36 |  **77.42**   | 70.51 |  **84.61**   | **70.98** |
>
> Performance of Different Filter Functions in Fine-Tune Setting. (F1 Score(%))
> |           |    AFQMC     |   CHIP-STS   | Chinese-QQP  |     MRPC     |    AVG    |
> | :-------: | :----------: | :----------: | :----------: | :----------: | :-------: |
> | Baseline  |    64.19     |    86.93     |    76.56     |    91.21     |   79.72   |
> |  Cosine   |    66.43     |    88.01     |    77.33     |    92.63     |   81.10   |
> |  Constant| 66.57 | 88.15 |    78.45     |    92.51     |   81.42   |
> | Quadratic |    66.37     |    87.76     |  **79.26**   | 92.75 |   81.54   |
> |  Linear   |  **66.59**   |  **88.39**   | 78.43 |  **92.78**   | **81.55** |
>
> We hope that our response has allayed the reviewer's concerns and any further concerns are appreciated.

---

### Official Review · Reviewer_6Yjm · 2022-10-27

**Confidence:** 3
**Correctness:** 3
**Technical Novelty And Significance:** 3
**Empirical Novelty And Significance:** 3
**Recommendation:** 6

**Clarity, Quality, Novelty And Reproducibility:**

The purpose and motivation were clearly explained, and is novel as providing a new way for training generator and discriminator with data selection. Authors claimed to open their code and data. Also with the explanation in the paper, the idea is not difficult to reproduce.

**Strength And Weaknesses:**

Strength: different from GAN where generator and discriminator are in a competitive learning strategy which is difficult to train, and difficult to use the generator data in further training (it is difficult to judge the quality of the generated data samples ), authors' idea with self-consistency learning could learn both the generator and discriminator in a cooperative way to enhance each other. Therefore, the training is stable and guarantee to converge.

Weakness: The dynamic threshold design as given in Eq. 3 seems important in deciding the selection of the data, however, authors gave a simple incremental formulation in later part for designing the dynamic threshold. And there is no further explanation and ablation study on the design of the dynamic threshold.

**Summary Of The Paper:**

In this study, authors proposed a self-consistent learning algorithm to generate data samples to improve downstream performance. The self-consistent learning was designed on generator-discriminator framework which is similar as GAN, but different from that of GAN, the generator and discriminator were cooperatively learned to enhance each other. In the self-consistent learning, authors designed a data selection mechanism to select the generated data with a dynamic threshold, and the selected data were further applied to enhance the generator and discriminator (similar as data augmentation). Finally, authors examined their idea on several sentence semantic matching experiments, and confirmed the effectiveness of their method.

**Summary Of The Review:**

The paper proposed a novel cooperative training of both generator and discriminator to generate data for enhance model training. In the model, authors designed a dynamic threshold for data generated data selection in a self-consistent learning framework. Authors experiments showed effectiveness of their idea.

---

> ### Author Response · Authors · 2022-11-15
> **Response to Reviewer 6Yjm**
>
> We thank the reviewer for the time and feedback. The comments are greatly appreciated and will be incorporated into a revised manuscript. We respond to specific questions and concerns below.
>
> q: there is no further explanation and ablation study on the design of the dynamic threshold.
> > We supplement Appendix B (EXPERIMENTS ON DIFFERENT FILTER FUNCTIONS) with further explanations and ablation experiments on different dynamic thresholding functions, including cosine, constant, quadratic, and linear functions.
>
> >As seen in Appendix B (listed below), compared to the baseline, these functions show different degree of improvement on the zero-shot and fine-tuning tasks, which indicate that dynmic thresholding is important, but the results are not so sensitive to the selection of the threshold functions. The linear function we use achieves the overall best performance across all tasks.
>
> Performance of Different Filter Functions in Zero-Shot Setting. (F1 Score(%))
> |           |    AFQMC     |   CHIP-STS   | Chinese-QQP  |     MRPC     |    AVG    |
> | :-------: | :----------: | :----------: | :----------: | :----------: | :---------: |
> | Baseline  |    38.25     |    58.82     |    57.88     |    68.54     |   55.87   |
> |  Cosine   |    47.38     | 74.26 |    64.39     |    83.48     |   67.38   |
> |  Constant   |    47.06     |    74.15     |    68.67     | 84.11 |   68.50   |
> | Quadratic |  **51.75**   |    73.09     |  **70.85**   |    83.48     |   69.79   |
> |  Linear   | 51.36 |  **77.42**   | 70.51 |  **84.61**   | **70.98** |
>
> Performance of Different Filter Functions in Fine-Tune Setting. (F1 Score(%))
> |           |    AFQMC     |   CHIP-STS   | Chinese-QQP  |     MRPC     |    AVG    |
> | :-------: | :----------: | :----------: | :----------: | :----------: | :-------: |
> | Baseline  |    64.19     |    86.93     |    76.56     |    91.21     |   79.72   |
> |  Cosine   |    66.43     |    88.01     |    77.33     |    92.63     |   81.10   |
> |  Constant   | 66.57 | 88.15 |    78.45     |    92.51     |   81.42   |
> | Quadratic |    66.37     |    87.76     |  **79.26**   | 92.75 |   81.54   |
> |  Linear   |  **66.59**   |  **88.39**   | 78.43 |  **92.78**   | **81.55** |
>
> Any further.concerns from the reviewer is appreciated.

---

> ### Comment · Reviewer_6Yjm · 2022-12-08
> **Authors added additional experiments to clear my questions.**
>
> I still keep the same score on the paper (rate 6)

---

### Official Review · Reviewer_r4Wp · 2022-11-03

**Confidence:** 3
**Correctness:** 3
**Technical Novelty And Significance:** 2
**Empirical Novelty And Significance:** 3
**Recommendation:** 6

**Clarity, Quality, Novelty And Reproducibility:**

In my opinion, the work is written in good quality and clear in general. Some questions mentioned above may need some clarifications. Codes were provided in supplementary materials, so it should be reproducible.

**Strength And Weaknesses:**

Strength: The paper is well-organized, and the main ideas are interesting, and easy to understand. The proposed method is evaluated on multiple datasets with good performance in sentence sematic matching.

Weakness: 1. The work included both positive and negative samples in the training of discriminator, and claimed cooperative setting can lead to better training stability than adversarial setting. The idea makes sense but it is not clear to me how this point is supported either in theoretical or experimental aspects. 2. The definition of the output probability for generator $p_G$ cannot be found. 3. It is mentioned in Eq (1) that $p_D$ depends on the embedding representation H of generated samples. Since it is shown in the paper that the selective mechanism is important to the performance, I think it would be helpful to provide more details on this embedding process, and some evidence or explanation to justify the resulting output probability is somehow “good enough” for selective mechanism. 4. It is mentioned in page 4 that both positive and negative samples would be included in the training of discriminator, (which is a key difference between this method and GAN.) Due to the lack of information on the embedding representation, I fail to see why Eq (2) can make sure of this.

**Summary Of The Paper:**

The work introduced a self-consistent learning framework, where a generator and a discriminator are trained cooperatively and alternatively. A selection mechanism is proposed to filter samples in the training process. The sentence semantic matching task is demonstrated as an example to support the effectiveness of the proposed framework. The results show the framework outperforms standalone discriminative models on four datasets in both zero-shot and full-data settings.

**Summary Of The Review:**

The paper includes interesting ideas and shows the proposed approach can improve performance for sentence sematic matching task. It may be more influential if fundamental evidences or analysis are available to compare cooperative and adversarial setting.

---

> ### Author Response · Authors · 2022-11-15
> **Response to Reviewer r4Wp**
>
> We thank the reviewer for taking the time to review our paper. we found the suggestions so insightful that we have supplemented them in the revised manuscript. We respond to specific questions and concerns below.
>
> q1: how this point is supported either in theoretical or experimental aspects?
>
> > To support our point that cooperative training can lead to better training stability and performance than the adversarial training in our closed-loop system, we conduct supplementary experiments on adversarial training. Specifically, we follow the idea of SeqGAN[1] by running Monte Carlo simulation several times at each token and  averaging the scores of the discriminator as the reward for that token. Then the generator is updated using  policy gradient method. Cross-entropy loss is used to train the discriminator, but all generated samples are treated as negative samples.
>
> > According to the experimental results in Appendix C, in all zero-shot settings, the performance of adversarial training is worse than the baseline, while for the fine-tuning case, the f1 scores of the discriminator on Chinese-QQP and MRPC datasets improve slightly compared to the baseline, but are still lower than that of cooperative training. It is worth noting that the entire process of adversarial training is extremely unstable, and fails to converge on most datasets.
>
> See Appendix C (CONTRASTIVE EXPERIMENTS WITH ADVERSARIAL TRAINING) in the manuscirpt for details.
>
> q2: The definition of the output probability for generator $p_G$ cannot be found.
> > We thank the reviewer for pointing out the lack of definition of $p_G$. Let us clarify it in more detail.  In classification tasks, for a given $k$-th class, $p_G$ refers to the degree of confidence that the sentences generated by the generator belongs to the class $k$. (We can either train the generator to express its confidence in the generated sentences [2] or use a fixed third-party model to score them [3]. Revisions have been maken accordingly in the main text.
>
> q3: provide more details on this embedding process, and some evidence or explanation to justify the resulting output probability is somehow “good enough” for selective mechanism.
> > We have provided an additional explanation of $\mathbf H$ in a footnote on page 3 of the paper. We follow the usage of Sentence-BERT[4] that takes the embedding representation of $CLS$-token as $\mathbf{H}$, and the input form to the discriminator is  "[CLS]sentence1[SEP]sentence2".
>
> > We empirically use different filter functions (including oscillating, fast-growing, constant, and linear) for the comparison. According to the experimental results in Appendix B, these functions show roughly the same degree of improvement on both the zero-shot and fine-tuning tasks compared to the baseline. In other words, we can get better performance without an overly tricky selection mechanism (i.e. threshold functions), which indicates that the resulting output probability is good enough for the selective mechanism.
>
> See Appendix B (EXPERIMENTS ON DIFFERENT FILTER FUNCTIONS) in the paper for details.
>
> q4: fail to see why Eq (2) can make sure of this.
> > We rewrite Eq (2) to make it more obvious. In the $t$-th round of training, for each class $k$, there is an threshold $\epsilon_{t,k}$ to filter samples that do not meet the conditions. For the 2-classes case, two thresholds would be used to select positive and negative samples respectively.
>
> We hope that the above explanations can answer the reviewer's questions. We appreciate the reviewer for raising further concerns to improve the manuscript.
>
> [1] Yu L, Zhang W, Wang J, et al. Seqgan: Sequence generative adversarial nets with policy gradient[C]//Proceedings of the AAAI conference on artificial intelligence. 2017, 31(1).
>
> [2] Lin S, Hilton J, Evans O. Teaching Models to Express Their Uncertainty in Words[J]. arXiv preprint arXiv:2205.14334, 2022.
>
> [3] Gao T, Yao X, Chen D. SimCSE: Simple Contrastive Learning of Sentence Embeddings[C]//Proceedings of the 2021 Conference on Empirical Methods in Natural Language Processing. 2021: 6894-6910.
>
> [4] Reimers N, Gurevych I. Sentence-BERT: Sentence Embeddings using Siamese BERT-Networks[C]//Proceedings of the 2019 Conference on Empirical Methods in Natural Language Processing and the 9th International Joint Conference on Natural Language Processing (EMNLP-IJCNLP). 2019: 3982-3992.

---

> > ### Comment · Reviewer_r4Wp · 2022-12-02
> > **Response to Authors**
> >
> > I want to thank the authors for their detailed responses. I think their explanations and the extra information in the revision are helpful. After reading and reviewing the contents, I would like to keep my rating as is.

---

### Author Response · Authors · 2022-11-15
**Response to All**

We would like to thank all of the reviewers for the valuable feedback and a detailed point-to-point response to the reviewers' comments will be given to each.

Regarding to the reviewers' concerns, we have added extensive discussions both in the main text and appendix, and all the changes made on the main text and appendix are colored in blue. Here we list the main points to polish the manuscript.
1. Improve the notation usage and add more detailed explanations to the proof of Theroem 1.
2. Ablation experiments on the selective mechanism of dynamic thresholds.
3. Add the results of adversarial training on all used datasets for comparison.

The supplementary experiments in points 2 and 3 reinforce our conclusion, that for NLP tasks in the self-consistent training framework we constructed, a closed loop formed by cooperation between the generator and the discriminator is far superior to the adversarial method. Also, dynamic thresholding is important, but the results are not so sensitive to the selection of the threshold functions.

---

### Decision · Program_Chairs · 2023-01-20

**Decision:**

Reject

**Justification For Why Not Higher Score:**

1. Reviewers agree that if the proposed algorithm is not completely new, then additional experiments on Vision should be conducted to further confirm the effectiveness of the proposed approach.
2. One reviewer provided several previous works [1-4] that share similar ideas to this paper.

[1] https://arxiv.org/pdf/1804.03782.pdf

[2] https://hal.archives-ouvertes.fr/hal-03736116/file/lamprier22a.pdf

[3] http://www.stat.ucla.edu/~ywu/PAMI.pdf

[4] https://www.sciencedirect.com/science/article/abs/pii/S0031320321002855

**Justification For Why Not Lower Score:**

1. The paper is well-written. The experimental results on text tasks are convincing.
2. The proposed algorithm could learn both the generator and discriminator in a cooperative way to enhance each other. Therefore, the training is stable and guaranteed to converge.

**Metareview: Summary, Strengths And Weaknesses:**

1. This paper proposes a self-consistent learning framework, in which a discriminator and a generator are cooperatively trained in a closed-loop form. Experimental results demonstrate the effectiveness of the proposed framework.
2. The proposed algorithm could learn both the generator and discriminator in a cooperative way to enhance each other. Therefore, the training is stable and guaranteed to converge.
3. The experiments only show the improvements on NLP tasks but not other tasks, such as Vision.
4. Several similar works (Cooperative Training for GAN) shave been proposed previously [1-4]. Thus, the novelty is relatively low.
5. The paper has some degree of scientific novelty, but may not meet ICLR's very high standards.

[1] https://arxiv.org/pdf/1804.03782.pdf

[2] https://hal.archives-ouvertes.fr/hal-03736116/file/lamprier22a.pdf

[3] http://www.stat.ucla.edu/~ywu/PAMI.pdf

[4] https://www.sciencedirect.com/science/article/abs/pii/S0031320321002855

**Summary Of Ac-Reviewer Meeting:**

1. Three reviewers maintained their scores (6, 6, 5) and one reviewer raised the score (from 1 to 3). All the reviewers agree that the paper has some degree of scientific novelty but may not meet the very high standard of ICLR.
2. Reviewers agree that if the proposed algorithm is not completely new, then additional experiments on Vision should be conducted to further confirm the effectiveness of the proposed approach.
3. One reviewer provided several previous works [1-4] that share similar ideas to this paper.

[1] https://arxiv.org/pdf/1804.03782.pdf

[2] https://hal.archives-ouvertes.fr/hal-03736116/file/lamprier22a.pdf

[3] http://www.stat.ucla.edu/~ywu/PAMI.pdf

[4] https://www.sciencedirect.com/science/article/abs/pii/S0031320321002855